# Evaluation of central bank independence, macroprudential policy, and credit gap in developing countries

Cep Jandi Anwar[1]*, Stephen G. Hall[2], Nermeen Harb[2], Indra Suhendra[1], Eka Purwanda[3]

1 Department of Economics and Development Studies, Faculty of Economics and Business, University of Sultan Ageng Tirtayasa, Kota Serang, Indonesia, 2 Division of Economics, University of Leicester School of Business, University of Leicester, Leicester, United Kingdom, 3 STIE STEMBI, Bandung Business School, Bandung, Indonesia

* cepjandianwar@untirta.ac.id

**Data Availability Statement:** All relevant data are within the paper and its Supporting Information files.

## Abstract

This study aims to examine whether Central Bank Independence (CBI) and Macroprudential Policy (MAPP) are capable of assisting the improvement of stability in the financial system, regarding the credit gap for 20 developing markets from 2000 to 2021. To examine this financial association, a panel threshold nonlinear model was implemented, based on the potentially time-varying influence of the CBI and MAPP index on the credit gap. The effects of this relationship also emphasized the CBI degree, whose greater level often stabilized the financial sector better. In this case, a stronger effect is commonly prioritized when CBI is below its trend. Based on the analysis, the selected experimental countries were categorized into two groups. The results showed that the nations with a higher CBI degree had greater stability in the financial system. Tighter MAPP also improved financial stability when CBI was below its trend. However, it did not enhance stability when CBI was more than the threshold level.

## 1. Introduction

Central Bank Independence (CBI) is highly critical in establishing macroeconomic stability, especially in controlling inflation levels. However, the global monetary crisis of 2007–2008 shifted its focus to stability in the financial system, which is a critical component of macroeconomic balance [1]. In reaction to the crisis, several central banks have raised financial stability to the same degree of priority as macroeconomic stability. This is based on claims that the credit cycle causes huge imbalances in credit and asset price bubbles, which can burst, resulting in severe recessions. Furthermore, central banks should utilise their monetary policy to keep credit and asset bubbles under control.

According to [2], CBI promoted financial system stability. This prioritized a high-level suggestion that central banks should avoid enacting pre-emptive monetary tightening, to guarantee stability. Several previous empirical studies, such as [3–5], examined the impact of CBI on financial system stability, regarding various indicators, including the soundness of banks and banking crisis likelihoods. Based on the results, more CBI was associated with a better stable

**Funding:** The author(s) received no specific funding for this work.

**Competing interests:** The authors have declared that no competing interests exist.

financial sector. To describe this sectoral stability, a state where the entire financial system is effectively operational is emphasized. Asides from the settlement and clearing systems, the financial sector also consists of fiscal institutions, markets, and payment methods.

CBI is fundamental for stability in the financial system for two main reasons. Firstly, the CBI is not subject to political pressure, indicating that the central bank has more latitude in preventing financial crises and formulating specific policies before materialization. When the bank is not independent, politicians often attempt to influence it, leading to delayed efforts to avert a fiscal catastrophe emphasizing potential conflicts of interest. Secondly, the financial stability policy has a time inconsistency issue. In this case, the policymaker needs a solution capable of assigning a CBI duty for upholding financial system stability. Thus, the effect of CBI on financial stability as objective of this study is important to be investigated.

In developing nations, a financial system is focused on banks [6–8], with a credit institution representing the most significant component of internal financing to a corporation within developing nations. Another reason prioritizes the capability of the institution in ensuring the greatest contribution to the financial system while being directly influenced by central bank policies. According to [5], financial stability was proxied by the likelihood of crises in the banking industry. This indicated that the use of crises to gauge fiscal soundness presented various difficulties. Firstly, banking crises are only discovered when they are sufficiently severe to cause market disruptions [4]. Secondly, the determination of the specific timing of a crisis is rather subjective, with its precision often queried [9]. Thirdly, banking crises are commonly analyzed only, leading to the negligence of the financial instability in other sectors. Since the credit gap is the strongest predictor of financial crises, this study employs it as a criterion of fiscal stability [10–12]. The loan gap was mostly used as a measure of financial stability by [13] because excessive credit expansion was a major factor in banking crises. Subsequently, [14, 15] analyzed this idea and concluded that historical credit gaps predicted future instability in the financial system.

Excessive credit growth is one of the main factors associated with banking or financial crises in developing countries. Even though an increase in credit can positively contribute to economic growth in the long run, in the short run it might lead to poor credit allocation creating economic imbalance [16]. Excessive credit growth can lead to financial crises via three channels: the first is by generating external macroeconomic imbalances; second, by inflating asset price resulting in bubbles and busts; third, by leading to inefficient use of resources [17]. The literatures differentiate credit to the private sector between household credit (consumption credit) and firm credit (productive credit). Household credit growth increases demand for consumption goods, which leads to an increase in consumption of goods and services.

Since most developing countries have low national saving rate, relaxation of the credit constraints creates an increase in household indebtedness without a similar rise in their future income, resulting in an increase in default risks [18]. Another consequence of a low saving rate is that credit growth is funded by international capital inflows which potentially increase financial crises [19]. On the other hand, an excessive growth in firm credit possibly causes banking crises via asset price bubbles. According to [20], growth in corporate credit results in higher leverage which can result in defaults if the firm or the economy experiences a major shock. This can create systemic defaults leading to banking crisis.

Excessive credit growth has long been seen as a major contributor to financial crises. The gap of credit-to-GDP ratio is a well-established metric for the degree to which credit expansion is excessive. According to the Basel Bank for International Settlements [21], definition of the gap is the indicator quantifies the deviations of the credit-to-GDP ratio from its trend as derived using a one-sided Hodrick-Prescott (HP) filter. The gap of credit-to-GDP ratio provides two purposes. First, it examines the amount of the economy's credit overhang. Second, it

causes the installation of countercyclical capital buffers in order to avoid a banking catastrophe caused by excessive loan expansion. As a result, this indicator is critical as an early warning signal of financial crises as well as a quantitative benchmark for macroprudential policy implementation.

Macroprudential Policy (MAPP) is one of the rregulations significantly affecting stability in the financial system. This policy is undertaken to reduce systemic risk, improve fiscal stability, and build a safer monetary system, toward the aversion of future crises. It also aims to limit the risk exposure of the financial system. Based on a few reports, a panel data regression was used with macro-level datasets, to assess the effectiveness of MAPP in lowering loan development. [22, 23] also examined the impact of MAPP on credit development, using cross-country data. The results showed that macroprudential policy was associated with less excessive credit growth.

Prior empirical studies which investigate the relationship between CBI and credit growth [4, 5, 24–26], macroprudential policy and credit growth [27–29] are performing the linear regression model and ignoring the possibility of a non-linear relationship. To investigate the relationship between CBI and credit gap, and macroprudential policy and credit gap, this paper constructs a non-linear methodology that captures the possible time-varying effect of CBI and macroprudential index on credit gap. Our hypotheses are the effects of CBI and macroprudential policy on credit gap is different when the degree of CBI is high and low. Thus, by performing a panel threshold non-linear regression, we can indicate what level of CBI is considered to be high and slow credit gap. This study also checks the homogeneity assumption of the threshold and coefficients of the parameters by performing a non-standard poolability test using the dummy variable approach.

The motivations of this study differs from the previous literature for three reasons. Firstly, a MAPP index is constructed by using the 12 Global Macroprudential Indicators (GMPI) of [22]. A new index is also developed through the coincident indicator method produced by [30, 31]. The 12 GMPI instruments are subsequently impacted by a common component, which is represented by an unobserved variable prioritizing one of the model's benefits. Secondly, the previous empirical reports emphasizing the relationship of CBI and MAPP with credit gap used a linear regression model to avoid the possibility of a non-linear association. To undertake a more in-depth study, a non-linear technique is developed, regarding the potentially time-varying influence of the CBI and MAPP indexes on the credit gap. The effects of these variables on the loan gap are also expected to change, regarding the degree of CBI. In this case, the CBI degree emphasizing high and sluggish credit gap is determined by using a panel threshold non-linear model. Thirdly, the Panel Smooth Transition Regression (PSTR) model of [32] assumes that the threshold and coefficients of parameters are homogenous across units. Based on this description, some differences are also observed for these criteria between nations. In this case, the dummy variable technique is used to perform a non-standard poolability test, toward validating homogeneity assumptions. Therefore, this study aims to examine whether CBI and MAPP are capable of assisting in the improvement of stability in the financial system, regarding the credit gap for 20 developing markets from 2000 to 2021.

The rest of this study is organised as follows, Section 2 is literature review, Section 3 emphasizes the implemented construction methodology and models, Section 4 explores the data set, Section 5 evaluates the empirical results, and Section 6 is the concluding session.

## 2. Literature review

This research examines the link between CBI, macroprudential policy, and financial stability using a theoretical model created by [33, 34]. They show that monetary policymakers raise

inflation when debt overhangs rise and macroprudential policy eases. Because the monetary authority will inflate some of the debt overhang, the macroprudential policymaker will have an incentive to create a looser policy that favours production and allows for further debt accumulation.

Numerous research indicate that CBI promotes financial stability [4, 5, 24–26]. [4] assess the impact of CBI on the likelihood of financial crises in 79 countries from 1970 to 1999. Using a logistic distribution model, they evaluate the impact of CBI on the incidence of a financial crisis. They conclude with evidence that CBI decreases the chance of banking crises considerably. [5] investigate the influence of CBI on bank soundness for data from 2000 to 2011 comprises 1,756 commercial banks from 94 countries. They demonstrate that CBI has a beneficial influence on bank soundness; hence, the higher the CBI score, the sounder the bank.

[24] examine the association between central bank independence and financial stability in 56 countries from 1980 to 2012. They discovered significant evidence linking central bank independence (personnel independence, financial independence, policy independence, and central bank objectives) to bank systemic risk. They suggest that the effect of central bank independence on systemic risk is greater during banking crises. Moreover, their results suggest that the democratic environment plays a vital role in moderating the central bank independence and systemic risk nexus. Their findings indicate that the democratic environment plays a crucial role in moderating the relationship between central bank independence and systemic risk. [25] investigate the relationship between the independent central bank and the systemic risk measures of banks. Their findings demonstrate that central bank independence has a negative, and significant effect on banks systemic risk. They conclude that an increase in central bank independence can mitigate the effects of environments characterised by a low level of financial freedom or high market power, which increase banks' contribution to systemic risk. [26] demonstrate that CBI weakens financial regulation, resulting in a variety of deregulatory approaches concerning financial reform, bank entry barriers, liberalisation, deposit insurance, and capital openness, and thus, negatively impact banking stability.

The empirical regarding macroprudential policy on financial stability including [13, 22, 27–29, 35, 36]. [27] offer empirical evidence from China regarding the influence of monetary policy and macroprudential policy coordination on financial stability and sustainability for the micro and macro level data from 2003 to 2017. They discovered that regulation of bank risk-taking, monetary policy, and macroprudential policy should all be counter-cyclical. [28] examine two sets of trade-offs, the first pertaining to the relative effectiveness of monetary and macroprudential policy in achieving price and financial stability objectives, and the second pertaining to the trade-offs between macroprudential policy instruments that enhance financial system resilience and those that aim to moderate the financial cycle. Their model demonstrates that macroprudential policies are more effective than monetary policy for bolstering the resilience of the financial system. Similarly, monetary policy is more effective at attaining price stability than macroprudential policy. [29] examine the effects of macroprudential and monetary policies on business cycles, economic well-being, and financial stability. They employ a dynamic stochastic general equilibrium (DSGE) model with constraints on housing and collateral. They conclude that given a positive technology or housing demand shock, the policymaker would reduce the macroprudential policy in order to moderate the credit growth, thereby achieving the financial stability.

[35] use quarterly data from 2000 to 2013 to assess how well macroprudential regulation controls loan growth in eleven emerging Asian nations. The qualitative vector autoregressive model generates the dynamic impulse response of credit growth to macroprudential policy changes for each sample nation. They believe macroprudential policy slows credit expansion. [22] examine how macroprudential regulation affected loan growth in 119 countries from

2000 to 2013. They find that macroprudential policy tightening reduces bank lending growth. After dividing the sample into developed, emerging, and developing nations, macroprudential regulation significantly slows credit growth in emerging and developing countries. This shows that emerging and developing nations reduce loan growth better with macroprudential policies.

[36] use quarterly data for 57 developed and emerging nations from 2000 to 2013 to assess the contribution of macroprudential policy. They demonstrate that a higher macroprudential policy index lowers the growth of total credit, the growth of housing credit, and the inflation of home prices. [13] investigate the impact of monetary and macroprudential policy on the macroeconomic variable for Australia, Indonesia, Korea, and Thailand, four inflation-targeting nations in the Asia Pacific region, from 2000 Q1 to 2014 Q4. Their study's major conclusion is that monetary and macroprudential measures considerably lower inflation and loan growth.

## 3. Econometric methodology

This article aims to explore the non-linear influence of CBI and its threshold on the credit gap. It also aims to investigate the influence of CBI, MAPP, and various control factors on the credit gap, when the central bank independence is above and below its threshold. Thus, our research questions are what are the relationship between CBI and credit gap, and macroprudential policy and credit gap?. Furthermore, what level of CBI is considered to be high and slow credit gap?. Our hypotheses are the effects of CBI and macroprudential policy on credit gap is different when the degree of CBI is high and low.

In this case, the PSTR of [32] was used to determine the CBI threshold level, the conversion pace between regimes, and the influence of explanatory factors on the low and high credit gap. However, some disadvantages were observed in the implementation of the PSTR method. Firstly, a heterogeneous CBI effect on the credit gap was not determined for all nations since the original PSTR model was a non-dynamic fixed effect model, where heterogeneity was reduced by the model's average. Secondly, the transition function ($\gamma$,c) differed between nations. Based on these descriptions, the PSTR model was implemented through panel non-linear least squares, assuming that the parameters and thresholds were consistent across nations. A non-standard poolability test was also used through dummy variables, to assess the pooling assumption of coefficients, explanatory indicators, and threshold. To validate the non-linearity hypothesis, the LR test was used to determine the non-linear model preferable to the linear method.

### 3.1 Panel threshold non-linear model

According to [32], PSTR estimation was a panel fixed effect model with exogenous regressors. Therefore, this model was implemented in a panel least squares model. The following is a formulation of a basic two-regime panel transition model,

$$y_{it} = \beta_0' x_{i,t} + [\beta_1' x_{i,t}] G(Z_{i,t;\gamma,c}) + \varepsilon_{it} \tag{1}$$

As a dependent variable, $y_{it}$ = the credit gap, $x_{i,t}$ = a vector of time-varying exogenous variables; $G(Z_{i,t};\gamma,c)$ = the transition function; and $\varepsilon_{it}$ = the error term. According to [20], the transition function was defined as $G(Z_{i,t};\gamma,c)$, which is a continuous expression of the observable variable, $Z_{i,t}$, normalized to be bounded between 0 and 1. For the low and high regimes, regression coefficients were $\beta_0'$ and $\beta_0' + \beta_1'$, respectively. The transition variable value $Z_{i,t}$ also determined the coefficient of $G(Z_{i,t};\gamma,c)$, indicating the regression expression was $\beta_0' + \beta_1' G(Z_{i,t;\gamma,c})$ for $i$ (country) at $t$ (time). [32] also proposed that the logistic specification function was

accompanied by the following transition function,

$$G(Z_{i,t;\gamma,c}) = (1 + \exp(-\gamma \prod_{j=1}^{m}(Z_{i,t} - c_j))) \text{ with } \gamma > 0 \text{ and } c_1 < c_2 < \ldots < c_m \tag{2}$$

Where $c = (c_1 \ldots c_m)^0$ is an $m$-dimensional vector of the location parameter, and $\gamma$ = the slope of smoothness between the low and high regimes.

In this model, only one transition function was assumably existent, with a location parameter also observed for the threshold variable. Therefore, this model emphasized an increase of $Z_{i,t}$, whose low and high values were separated by two different regimes, $\beta_0$ and $\beta_0 + \beta_1$, with a single monotonic transition of the coefficient. When $\gamma$ is the high value, the logistic transition, $G(Z_{i,t};\gamma,c)$, becomes the indicator function, $G(Z_{i,t},c)$. For $\gamma \to \infty$, the indicator function, $G(Z_{i,t}, c) = 1$ and 0 when $Z_{i,t} > c$ and $Z_{i,t} < c$, respectively. When $\gamma \to 0$, the transition function, $G(Z_{i,t};\gamma,c)$, was then constant, with Eq 1 becoming a linear model.

### 3.2 Linearity test

To prevent an unidentified model, the non-linear method was compared to a broader alternative (the linear model). This analysis assumed the false expression of $H_0 = 0$ or $H_0 = \beta_0 = \beta_1$. Under $H_0$, the nonlinear model contained unnamed nuisance parameters, leading to the non-standard nature of the test. Following [32, 37] addressed the problem by substituting $G(Z_{i,t};\gamma, c)$ in Eq (1), with its first-order Taylor expansion rounded to 0. Eq (1) was then modified by the auxiliary regression to produce the following expression,

$$y_{it} = \beta_0' x_{i,t} + \beta_1' x_{i,t} Z_{i,t} + \cdots + \beta_1' x_{i,t} Z_{i,t}^m + \varepsilon_{it} \tag{3}$$

This article examined credit development as a function of the CBI and MAPP indexes, inflation, economic growth, as well as interest and currency exchange rates. Moreover, the transition variable was CBI, with the nonlinear threshold model for the two-regime credit growth expressed as follows,

$$\begin{aligned}
Credit_{it} = {}& \beta_0 + \beta_{01}CBI_{it} + \beta_{02}MaPP_{it} + \beta_{03}Inflation_{it} + \beta_{04}Growth_{it} + \beta_{05}IR_{it} + \beta_{06}ER_{it} \\
& + (\beta_{11}CBI_{it} + \beta_{12}MaPP_{it} + \beta_{13}Inflation_{it} + \beta_{14}Growth_{it} + \beta_{15}IR_{it} \\
& + \beta_{16}ER_{it})G(CBI_{i,t;\gamma,c}) + \varepsilon_{it}
\end{aligned} \tag{4}$$

By substituting $G(Z_{i,t};\gamma,c)$ with the first-order Taylor expansion in Eq (4), 0 was obtained. This equation then became a linear model by using the auxiliary regression, as shown in the following expression.

$$\begin{aligned}
Credit_{it} = {}& \beta_0 + \beta_{01}CBI_{it} + \beta_{02}MaPP_{it} + \beta_{03}Inflation_{it} + \beta_{04}Growth_{it} + \beta_{05}IR_{it} + \beta_{06}ER_{it} \\
& + \varepsilon_{it}
\end{aligned} \tag{5}$$

Under the null hypothesis, the test was conducted by using the likelihood ratio test, whose distribution was $2\chi^2(df)$. The test for this ratio is then observed as follows,

$$LR\ Test = 2*[l_U - l_R] \tag{6}$$

Where $l_R$ = the log-probability under $H_0$ (linear model) and $l_U$ = the log-probability under $H_1$ (non-linear model).

### 3.3 Poolability test

To determine the poolability of the model, a non-standard test was carried out, with the restricted and unrestricted methods then compared. Since Eq 1 showed the unrestricted model, a dummy variable approach was used, where D = 0. Meanwhile, D = 1 for i (country)

in the restricted model. Each dependent variable was also multiplied by the dummy indicator, with the two connected criteria (0 and 1) permitting the slope to change as follows,

$$y_{it} = \beta'_0 \, x_{i,t} + [\beta'_1 \, x_{i,t}]G(Z_{i,t;\gamma,c}) + [\beta'_0 \, x_{i,t} + [\beta'_1 \, x_{i,t}]G(Z_{i,t;\gamma,c})]*Dummy + \varepsilon_{it} \qquad (7)$$

## 4. Data

The analyzed panel data included 20 developing nations which have changes in the CBI and MAPP indexes from 2000 to 2021. The number of cross-sections is also affected by the availability of data for other possible factors. To document the importance of the MAPP index, the model GMPI model constructed by [22] was implemented. In this case, 1 was assigned to each instrument, with 0 observed otherwise. Since the GMPI index is the sum of the score for all 12 tools, a downside is then observed in combining all 12 instruments. This method is highly unweighted due to all instruments using identical weights. In this study, a new index was developed based on the Dynamic Factor Model (DFM) published by [30, 31].

Regarding the legal component to measure central bank independence, the CBI index was used. This index ranged from 0 to 1, with higher values indicating greater CBI. The data collection method of [38] was also used to derive the CBI index information. Furthermore, the credit gap is defined as the difference between the aggregate credit-to-GDP ratio and its long-run trend. This ratio long-run trend was calculated by using the Hodrick-Prescott (HP) filter. In this analysis, the implemented private credit per GDP emphasized the entire debt of families and domestic non-financial sectors related to the financial institution in proportion to GDP. The data were also obtained from the World Bank's Global Financial Development Database. Extensive reports subsequently demonstrated that a significant credit gap was very reliable for financial crises [39].

Economic growth, inflation, as well as currency and interest rates were included in the analysis as control variables. According to [40], inflation significantly affected the credit gap. This was because inflation was the percentage change in the consumer price index across the relevant period. The priori sign of inflation on credit gap was also positive, indicating its detrimental nature and uncertainty to financial system stability. Moreover, economic development is the primary determinant of the credit gap, with the GDP growth rate being its measure. According to [3], stronger economic development should minimise the credit gap when public income is considerable. This indicated the needlessness to borrow money from a bank. The interest rate is also assessed as the policy rate of the central bank and is derived from the IMF's IFS. In this case, a high-interest rate often reduces bank credit demand. For the exchange rate, the value between the US Dollar and the local currency is bilateral. Based on [41], the primary element in the credit gap was the currency exchange rate, whose statistics were commonly obtained from the IMF's IFS.

## 5. Empirical results

### 5.1 Linearity test

To evaluate whether the non-linear model was preferable to the linear type, a log-likelihood ratio test was conducted. In this analysis, the non-linear ($l_U$) and linear ($l_R$) models were -3921.09 and -3943.85, respectively. The computed LR ratio was also 45.52, while the $\chi^2(10)$ with 5% significance was 18.31. Since the LR ratio was greater than the $\chi^2$ statistic, the null hypothesis was rejected at the 5% significance level. This demonstrated that the presence of a nonlinear relationship between CBI and credit gap significantly outperformed the linear model.

## 5.2 Panel threshold non-linear model

Table 1 showed the outcome of the nonlinear model with a CBI threshold. This indicated that the calculated CBI threshold was 0.3545, demonstrating the existence of a low regime when the central bank independence degree was less than or equal to 0.3545. Meanwhile, a high regime was observed when the CBI index was more than 0.1299. In this case, the slope of the transition between regimes was 6.8611.

Based on the results, CBI greatly reduced the credit gap in the high regime, although it was ineffective in the low phase. This showed that the strength of the CBI was highly effective in greatly reducing the loan gap. However, neither the low nor high regimes showed a substantial influence of macroprudential regulation on this gap. In the low regime, inflation and exchange rate positively and considerably affected the credit gap, although the economic growth and interest values reduced it significantly. For the high regime, economic growth and the exchange rate narrowed the gap, with the interest rate considerably widening it.

**5.2.1 Poolability test.** According to the results, the potential for sample heterogeneity was neglected, with the assumption that all explanatory variable coefficients were uniform across all countries. To determine whether the coefficients for explanatory variables were pooled, the pooling assumption in the model was evaluated.

In Table 2, the poolability test was carried out by using the dummy variable strategy. This showed that 10 of the 20 countries in our sample were pooled, with the remaining ten unused. Based on these results, the assumption of homogeneity for the whole sample was inapplicable. This explained that the coefficients were different for each country, indicating the existence of heterogeneity in the model and the biased conclusion within Table 1. Regarding the poolability test, the samples were then categorized into two divisions, namely Groups 1 and 2, which contained poolable and non-poolable countries, respectively.

**5.2.2 Split sample.** Table 3 considers two subgroups, namely Groups 1 and 2, which contained poolable and non-poolable countries, regarding the poolability test. In this analysis, the first group contained Algeria, Argentina, China, Djibouti, Dominican Republic, Egypt, Indonesia, Kenya, Sri Lanka and Tanzania. However, the second group encompassed Belarus, Bhutan, Kazakhstan, Malaysia, Maldives, Nepal, Nicaragua, Morocco, Thailand, and South Africa.

In columns 2 and 3 of Table 3, the estimated coefficient of the panel threshold model for Group 1 was displayed. This indicated that the transition and threshold values were 4.8926 and 0.4838, respectively. From these results, CBI negatively and substantially affected the credit

**Table 1. Estimation result of non-linear model.**

| Variable | High Regime | Low Regime |
|---|---|---|
| CBI | -4.8926*** (1.4428) | -3.5234 (2.7706) |
| MAPP | -1.2173 (0.9907) | -6.4631 (-4.5104) |
| Inflation | -0.0035 (0.0182) | 0.0357*** (0.0091) |
| Growth | -0.0279*** (0.0039) | -0.2246** (0.1074) |
| Interest Rate | 0.0664*** (0.0129) | -0.0717*** (0.0246) |
| Exchange Rate | -0.1070*** (0.0053) | 0.4494** (0.2164) |
| Threshold (c) | 0.3545*** | |
| Slope ($\gamma$) | 6.8611*** | |

Credit gap as a dependent variable.

* is Prob. <10%

** is Prob. <5%, and

*** is Prob. <1%.

**Table 2. Poolability test.**

| Country | lU | lR | LR Statistic | χ2 | Summary |
|---|---|---|---|---|---|
| Algeria | -3,918.69 | -3,921.09 | 4.802 | 21.03 | Poolable |
| Argentina | -3,914.88 | -3,921.09 | 12.428 | 21.03 | Poolable |
| China | -3,921.12 | -3,921.09 | -0.052 | 21.03 | Poolable |
| Belarus | -3,860.36 | -3,921.09 | 121.458 | 21.03 | Not Poolable |
| Bhutan | -3,910.18 | -3,921.09 | 21.828 | 21.03 | Not Poolable |
| Djibouti | -3,912.64 | -3,921.09 | 16.894 | 21.03 | Poolable |
| Dominican Rep | -3,914.45 | -3,921.09 | 13.28 | 21.03 | Poolable |
| Egypt | -3,914.49 | -3,921.09 | 13.202 | 21.03 | Poolable |
| Indonesia | -3,916.58 | -3,921.09 | 9.01 | 21.03 | Poolable |
| Kazakhstan | -3,894.73 | -3,921.09 | 52.714 | 21.03 | Not Poolable |
| Kenya | -3,917.27 | -3,921.09 | 7.632 | 21.03 | Poolable |
| Malaysia | -3,904.16 | -3,921.09 | 33.86 | 21.03 | Not Poolable |
| Maldives | -3,887.68 | -3,921.09 | 66.824 | 21.03 | Not Poolable |
| Nepal | -3,898.79 | -3,921.09 | 44.606 | 21.03 | Not Poolable |
| Nicaragua | -3,903.53 | -3,921.09 | 35.112 | 21.03 | Not Poolable |
| Morocco | -3,900.83 | -3,921.09 | 40.516 | 21.03 | Not Poolable |
| Thailand | -3,888.33 | -3,921.09 | 65.51 | 21.03 | Not Poolable |
| South Africa | -3,826.79 | -3,921.09 | 188.606 | 21.03 | Not Poolable |
| Sri Lanka | -3,915.73 | -3,921.09 | 10.722 | 21.03 | Poolable |
| Tanzania | -3,920.81 | -3,921.09 | 0.554 | 21.03 | Poolable |

gap, with -31.9264 and -6.8611 observed for low and high regimes, respectively. In this case, CBI development emphasized monetary establishment, which decreased the credit gap ratio and the likelihood of financial system instability.

For the MAPP index, a considerable negative influence was observed on the credit gap when CBI exceeded the threshold value. When the central bank was extremely independent, MAPP tightening was effective in narrowing the gap. However, when the CBI was below the threshold level, MAPP positively and substantially affected the credit gap, as measured by a coefficient of 6.4631. This proved that a 1% point rise in the MAPP index led to a 0.06%

**Table 3. Split sample group.**

| Variable | Group 1 | | Group 2 | |
|---|---|---|---|---|
| | **High Regime** | **Low Regime** | **High Regime** | **Low Regime** |
| CBI | -6.8611*** (0.7154) | -31.9264*** (9.5993) | -4.9468*** (0.9396) | -37.3422*** (11.2874) |
| MAPP | -1.2173*** (0.1631) | 6.4631*** (3.4009) | 3.2968*** (1.3191) | -10.2935*** (3.7016) |
| Inflation | -0.0036*** (0.0008) | 0.0357*** (0.0125) | 0.0936 (0.1044) | 0.0411** (0.0187) |
| Growth | -0.2279 (0.2053) | 0.2246 (0.2658) | 0.1426 (0.1656) | -0.1611** (0.0817) |
| Interest Rate | 0.0664*** (0.0149) | -0.0717*** (0.0167) | 0.0207*** (0.0063) | -0.2097*** (0.0589) |
| Exchange Rate | -0.1070*** (0.0105) | 0.4494 (0.4572) | -0.2817*** (0.0385) | 0.1974 (0.1332) |
| Threshold (c) | 0.4838*** | | 0.5306*** | |
| Slope (γ) | 4.8926*** | | 4.6298*** | |

Note: Credit gap as a dependent variable.

* is Prob. <10%

** is Prob. <5%, and

*** is Prob. <1%.

increase within the gap. One possible reason for this positive relationship emphasized the application of MAPP in numerous countries for many years, although it was highly considered after the global financial crisis [42]. For the low regime, the MAPP index occurred before the credit boom, indicating the requirement for an increased loan gap to stimulate investment.

In columns 4–5 of Table 3, the outcome of the panel threshold non-linear model for the non-poolable group was observed. For the transition velocity, the predicted slope parameter from the low to the high regime was 4.6298. Meanwhile, the estimated location parameter of CBI was 0.5306. From these results, CBI issued a lower credit gap, although the effect was only substantial for the high regime. This indicated that a greater CBI was successful in closing the gap. At -10.2935 for the low regime, MAPP also negatively and considerably influenced the credit gap. In contrast, increased MAPP dramatically widened this gap in the high regime.

**5.2.3 Mean group estimator of full sample.** For the overall sample of countries, the mean group sizes were estimated. In this analysis, a weighted average was adopted for the coefficients of groups 1 and 2 (poolable and non-poolable countries), regarding the number of countries in each category.

Based on Table 4, the CBI threshold level was 0.5072, with the transition between the low and high regimes being 4.7612 smooth. This indicated that CBI negatively affected the credit gap between the lowest and maximum threshold. In the low regime, this negative impact was greater than in the high regime. This proved that a 1%-point rise in CBI decreased the credit gap in the low and high regimes by approximately 0.34% and 0.06%, respectively. This negative correlation confirmed that CBI was successful in reducing financial instability, and was consistent with the results of [2, 4, 5, 24–26, 43]. According to [44], financial stability was achieved after the price balance was secured. This showed that CBI had a substantial moderating influence on lending growth.

Regarding the credit gap, the impact of MAPP varied among regimes, with positivity and negativity observed in the high and low sectors. For the low regime, a 1% increase in the MAPP index decreased the gap by approximately 0.19%. This result was confirmed by [22, 23], where a negative relationship was found between MAPP and credit gap. At 1.0299, MAPP positively influenced the gap when the CBI index was above the minimal level. This indicated that a 1% rise in the MAPP index elevated the credit gap by 0.1039% when CBI was high. The result was then corroborated by [13], where CBI and MAPP were synergistic for achieving financial and price stability during normal times. However, when inflation was low and stable with a

**Table 4. Mean Group Estimation (MGE).**

| Variable | High Regime | Low Regime |
|---|---|---|
| CBI | -5.9039*** (0.8275) | -34.6343*** (10.4433) |
| MAPP | 1.0397*** (0.5780) | -1.9152*** (0.1503) |
| Inflation | 0.0045 (0.0518) | 0.0384*** (0.0156) |
| Growth | -0.0426 (0.0794) | 0.0317 (0.0920) |
| Interest Rate | 0.0435*** (0.0106) | -0.1407*** (0.0378) |
| Exchange Rate | -0.1943*** (0.0245) | 0.3234 (0.2952) |
| Threshold (c) | 0.5072*** | |
| Slope ($\gamma$) | 4.7612 | |

Credit gap as a dependent variable.

* is Prob. <10%

** is Prob. <5%, and

*** is Prob. <1%.

**Table 5. Comparison sub-sample group.**

| Variable | MG-Full Sample | | Group 1 | | Group 2 | |
|---|---|---|---|---|---|---|
| | High Regime | Low Regime | High Regime | Low Regime | High Regime | Low Regime |
| CBI | -5.9039*** (0.8275) | -34.6343*** (10.4433) | -6.8611*** (0.7154) | -31.9264*** (9.5993) | -4.9468*** (0.9396) | -37.3422*** (11.2874) |
| MAPP | 1.0397*** (0.5780) | -1.9152*** (0.1503) | -1.2173*** (0.1631) | 6.4631*** (3.4009) | 3.2968*** (1.3191) | -10.2935*** (3.7016) |
| Inflation | 0.0045 (0.0518) | 0.0384*** (0.0156) | -0.0036*** (0.0008) | 0.0357*** (0.0125) | 0.0936 (0.1044) | 0.0411** (0.0187) |
| Growth | -0.0426 (0.0794) | 0.0317 (0.0920) | -0.2279 (0.2053) | 0.2246 (0.2658) | 0.1426 (0.1656) | -0.1611** (0.0817) |
| Interest Rate | 0.0435*** (0.0106) | -0.1407*** (0.0378) | 0.0664*** (0.0149) | -0.0717*** (0.0167) | 0.0207*** (0.0063) | -0.2097*** (0.0589) |
| Exchange Rate | -0.1943*** (0.0245) | 0.3234 (0.2952) | -0.1070*** (0.0105) | 0.4494 (0.4572) | -0.2817*** (0.0385) | 0.1974 (0.1332) |
| Threshold (c) | 0.5072*** | | 0.4838*** | | 0.5306*** | |
| Slope ($\gamma$) | 4.7612 | | 4.8926*** | | 4.6298*** | |

Note: Credit gap as a dependent variable.

\* is Prob. <10%

\*\* is Prob. <5%, and

\*\*\* is Prob. <1%.

substantial credit gap, the monetary policymaker had a dilemma due to the existence of a trade-off between financial and price stability. Higher CBI and subsequent MAPP tightening also stabilized one aim only, although their implementation in the other direction caused the policies serving distinct ends.

**5.2.4 Comparison sub-sample group.** In this section, the impacts of CBI, MAPP, inflation, economic growth, as well as interest and currency rates on the credit gap were analyzed in two distinct categories, namely Groups 1 and 2. Since this is the mean of both groups, the coefficients were constantly distributed. Based on the results, the model's sensitivity to various factors was found. These factors included the exchange rate regime, CBI degree, and monetary policy tightening.

Table 5 illustrates panel threshold nonlinear regression for two distinct groups and the overall sample mean. Interestingly, the transition threshold and rate varied between groups 1 and 2. This demonstrated that the CBI threshold levels for Groups 1 and 2 were 0.4838 and 0.5306, respectively. Meanwhile, the average CBI threshold level for all samples was 0.5072. The rates of change between low and high regimes were also 4.8926 and 4.6298 for Groups 1 and 2, respectively. The average rate for the entire sample was also 4.7612. From these results, Group 1 had a lower threshold and slope than Group 2, due to the differing CBI degrees in each category.

For all categories in both low and high regimes, a negative correlation was also observed between CBI and the credit gap. This showed that CBI promoted stability in the financial system despite being below or over the threshold. In the low regime, the coefficients of Groups 1 and 2, as well as the sample mean were -31.9264, -37.3422, and -34.6343, respectively. However, the coefficients of -6.8611, -4.9468, and -5.9039 were observed for Groups 1 and 2, as well as the sample mean in the high regime, respectively. From these results, the adverse impact of CBI on the credit gap was greater in Group 1 than in Group 2. This greater effect emphasized the output of a larger CBI degree in Group 2. In this case, the CBI indexes in Groups 1 and 2 were above and below average, respectively. These results were consistent with [4], where the nations with an above-average CBI degree had more financial system stability than those with a below-average level.

Based on the MAPP index, Groups 1 and 2 showed considerably different outputs in the low and high regimes. In Group 1, the effect of macroprudential regulation on the credit gap was favourable and negative under low and high regime conditions, respectively. Meanwhile,

the relationship was negative and positive in the low and high regimes within Group 2, respectively. From these results, no substantial difference was found between both groups, regarding the MAPP index. A difference was, however, observed in the credit gap between the two divisions, with Group 1 found to be lower than Group 2 at 0.0113 and 0.1610, respectively. When the degree of CBI was minimal in this group, stricter MAPP outputs were also observed in a smaller credit gap. This was because policymakers employed macroprudential devices to narrow the credit gap, regarding their awareness of financial instability. The interplay between a larger CBI and a more stringent macroprudential regulation also led to financial instability. This was due to the implementation of ineffective measures by the central bank when it has two policy objectives, namely financial and price stability [27–29, 45–47]. At a high CBI index for Group 1, stronger macroprudential regulation reduced the credit gap. This provided credence to the notion that a higher MAPP index enhanced financial system stability, especially in a central bank with a high degree of independence.

**5.2.5 Robustness test.** The robustness of the CBI effect on the credit gap was evaluated through Instrumental Variable (IV) estimates. In this case, the System Generalized Method of Moments (Sys GMM) Estimator by [48] was implemented, to obtain potential endogeneity. The CBI was endogenously treated and represented by lagging numbers in the first difference equation.

According to Table 6, the p-values AR (1) and AR (2) were significant and insignificant, respectively. This demonstrated the consistency of the Sys GMM estimator. Regarding the Sargan test, the validity of the set of instrumental variables was also verified. This proved that the instruments were not associated with the residuals, indicating the null hypothesis of the Hansen test. From these results, the Sargan test stated that the null hypothesis was not rejected, demonstrating the validity of the estimator using the Sys GMM approach. This approach was found to satisfy the requirements for the suitable panel IV model, concerning the outputs of the AR (1), AR (2), and Sargan tests.

Table 6 also showed that CBI and MAPP negatively and significantly affected the credit gap in the three groups. This result was consistent with the primary estimation method, which

**Table 6. Panel Sys GMM estimation.**

| Variable | Full Sample | Group 1 | Group 2 |
|---|---|---|---|
| Credit Gap (-1) | -0.5731*** (0.1119) | -0.6970*** (0.0213) | -0.3730*** (0.0170) |
| CBI | -2.5203*** (0.4683) | -2.3327*** (0.8233) | -2.8546*** (1.0027) |
| MAPP | -1.9593*** (0.6129) | -1.2707*** (0.4186) | -1.7954*** (0.5128) |
| Inflation | -0.0192** (0.0017) | -0.0227*** (0.0004) | -0.4582*** (0.0005) |
| Growth | -0.0200** (0.0094) | 0.0119*** (0.0018) | -0.0458*** (0.0095) |
| Interest Rate | -0.0117*** (0.0037) | -0.0034*** (0.0009) | -0.0660*** (0.0014) |
| Exchange Rate | 1.7929*** (0.5393) | 1.0314*** (0.0952) | 0.0037*** (0.0013) |
| AR (1) (p-value) | 0.0086 | 0.0097 | 0.0093 |
| AR (2) (p-value) | 0.2614 | 0.2439 | 0.6048 |
| Sargan Test (p-value) | 1.0000 | 1.0000 | 1.0000 |
| No. of Cross-section | 20 | 20 | 20 |
| No. of Observation | 1640 | 820 | 820 |

Note: The table reports coefficients from panel instrumental variable estimation. Credit gap as a dependent variable.

* is Prob. <10%

** is Prob. <5%, and

*** is Prob. <1%.

used a non-linear approach for the full and separated samples. Therefore, CBI and MAPP played crucial roles in reducing the credit gap within developing markets.

## 6. Conclusion

Based on these results, financial and price stability were synergized because of CBI. This indicated that a central bank with greater independence was advantageous for narrowing the credit gap, leading to the reduction of instability in the financial system. The results were also consistent with [43]. Similarly, these data reliably validated the conclusion reached by [4], where CBI was beneficial in mitigating financial instability. When the sample was divided into two groups based on the poolability test, a negative correlation was also observed between CBI and the credit-to-GDP ratio. The size of the effect was subsequently greater for the poolable group, due to the higher-than-average CBI degree, compared to the non-poolable category. In the low regime, CBI also greatly affected the credit gap than in the high sector. This was due to a modest CBI shift in the low regime, leading to a greater marginal effect. When the CBI degree was below and above its threshold, the macroprudential intervention had a varied influence on the credit gap. In this case, MAPP promoted financial stability for the low regime. However, a trade-off was observed between pricing and financial stability when the degree of CBI was large. An increase in CBI and a tightening of MAPP also stabilized one aim only. This indicated significant policy implications, especially for emerging nations with a low financial development index. From this context, these nations need to strengthen their central banks and minimize financial instability through stricter macroprudential regulation.

This study suggests that CBI may promote financial stability. Low development financial institutions have lower credit with greater CBI and tighter macroprudential policies. For nations with high-quality financial institutions, a smaller CBI and stronger macroprudential policies reduce credit gap. The results show that CBI and macroprudential policy are substitutes, not complements. This conclusion has substantial policy implications, especially for developing nations with a low financial development index that need to strengthen their central bank and tighten macroprudential policies to avoid financial instability. The implications of alternative metrics of financial stability based on bank credit, such as the amount of Non-Performing Loans (NPL) and bank soundness, may be the subject of future research. In a subsequent research, the threshold effect on the macroprudential policy index may also be estimated. It would be interesting to assess whether there is an ideal level of macroprudential policy index that alters the effect of macroprudential policy on loan growth.

## Supporting information

**S1 Data.**
(PDF)

## Author Contributions

**Conceptualization:** Cep Jandi Anwar, Stephen G. Hall.

**Data curation:** Cep Jandi Anwar.

**Formal analysis:** Indra Suhendra, Eka Purwanda.

**Investigation:** Eka Purwanda.

**Methodology:** Cep Jandi Anwar, Stephen G. Hall, Nermeen Harb.

**Resources:** Indra Suhendra.

**Software:** Nermeen Harb.

**Writing – original draft:** Cep Jandi Anwar.

**Writing – review & editing:** Cep Jandi Anwar.

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
