## [Decision Letter · Decision Letter 0]

30 Jan 2023

PONE-D-23-00739Evaluation of Central Bank Independence, Macroprudential Policy, and Credit Gap in Developing CountriesPLOS ONE

Dear Dr. Anwar,

Thank you for submitting your manuscript to PLOS ONE. After careful consideration, we feel that it has merit but does not fully meet PLOS ONE’s publication criteria as it currently stands. Therefore, we invite you to submit a revised version of the manuscript that addresses the points raised during the review process.

We look forward to receiving your revised manuscript.

Kind regards,

Muhammad Kamran Khan, PhD Finance

Academic Editor

PLOS ONE

Journal Requirements:

Additional Editor Comments:

1. The paper’s introduction section is not well organized therefore not well written at all. Too much reference has been cited in the introduction section which makes the reading uninteresting. There is hardly any linking conjunction, adverbs, propositions used in the whole manuscript which makes the paper incoherent. It seems that some references have been picked up from difference sources and jumbled up together. It’s also obsolete to use past tenses while stating a reference. There are no concluding sentences at the end of each paragraph of the introduction section. So, the introduction must be rewritten.

2. The proof reading has not been done professionally. A lot of grammatical mistakes are also evident.

3. The authors must clearly state why this study is important to conduct in the introduction section. The objective of the study is not mentioned in the introduction.

4. The contribution of the study is very poorly stated in two sentences and shockingly understated and disorganized.

5. It is utterly confusing the purpose of the literature mentioned in the introduction. Are they to mention the shortcomings of the existing literature or for mentioning the importance of conducting this study?

6. The gap or shortcomings of the literature has not been mentioned anywhere in the literature review section. The authors continue to describe different studies but did not mention the gap in the literature. This section must be rewritten as well. It’s really hard to understand what authors mean by mentioning these studies.

7. The result and discussion section are very much disorganized. The authors should clearly state which studies are in line with their findings and which contradicts.

8. The paper does not mention any theoretical or anecdotal evidences which actually justify the selection of the variables in the context of used economies.

9. The robustness of the results has not been checked. The robustness can be checked by changing model idealizations or changing background conditions.

Reviewers' comments:

Reviewer's Responses to Questions

**Comments to the Author**

1. Is the manuscript technically sound, and do the data support the conclusions?

Reviewer #1: Yes

2. Has the statistical analysis been performed appropriately and rigorously? 

Reviewer #1: Yes

3. Have the authors made all data underlying the findings in their manuscript fully available?

Reviewer #1: Yes

4. Is the manuscript presented in an intelligible fashion and written in standard English?

Reviewer #1: Yes

5. Review Comments to the Author

Reviewer #1: 1.A stronger motivation and economic intuition for the topic should be provided in introduction section. Scarcity does not mean necessity. You need to prove your paper is valuable enough and distinguish from others.

2. Separate the literature review from introduction section.

3. Based upon your empirical results, propose concrete policy recommendations and provide future directions for researchers.

6. PLOS authors have the option to publish the peer review history of their article (what does this mean?). If published, this will include your full peer review and any attached files.

Reviewer #1: No

While revising your submission, please upload your figure files to the Preflight Analysis and Conversion Engine (PACE) digital diagnostic tool, https://pacev2.apexcovantage.com/. PACE helps ensure that figures meet PLOS requirements. To use PACE, you must first register as a user. Registration is free. Then, login and navigate to the UPLOAD tab, where you will find detailed instructions on how to use the tool. If you encounter any issues or have any questions when using PACE, please email PLOS at figures@plos.org. Please note that Supporting Information files do not need this step.<quillbot-extension-portal></quillbot-extension-portal>

---

## [Author Response · Author response to Decision Letter 0]

24 Mar 2023

I provide a letter of response to reviewer's

---

## [Decision Letter · Decision Letter 1]

6 Apr 2023

PONE-D-23-00739R1Evaluation of Central Bank Independence, Macroprudential Policy, and Credit Gap in Developing CountriesPLOS ONE

Dear Dr. ANWAR,

Thank you for submitting your manuscript to PLOS ONE. After careful consideration, we feel that it has merit but does not fully meet PLOS ONE’s publication criteria as it currently stands. Therefore, we invite you to submit a revised version of the manuscript that addresses the points raised during the review process.

We look forward to receiving your revised manuscript.

Kind regards,

Muhammad Kamran Khan, PhD Finance

Academic Editor

PLOS ONE

Journal Requirements:

Reviewers' comments:

Reviewer's Responses to Questions

**Comments to the Author**

1. If the authors have adequately addressed your comments raised in a previous round of review and you feel that this manuscript is now acceptable for publication, you may indicate that here to bypass the “Comments to the Author” section, enter your conflict of interest statement in the “Confidential to Editor” section, and submit your "Accept" recommendation.

Reviewer #1: (No Response)

Reviewer #2: (No Response)

2. Is the manuscript technically sound, and do the data support the conclusions?

Reviewer #1: (No Response)

Reviewer #2: Partly

3. Has the statistical analysis been performed appropriately and rigorously? 

Reviewer #1: (No Response)

Reviewer #2: Yes

4. Have the authors made all data underlying the findings in their manuscript fully available?

Reviewer #1: (No Response)

Reviewer #2: Yes

5. Is the manuscript presented in an intelligible fashion and written in standard English?

Reviewer #1: (No Response)

Reviewer #2: Yes

6. Review Comments to the Author

Reviewer #1: (No Response)

Reviewer #2: 1. Title is adequate

2. it would be helpful to provide more context about the credit gap and its significance in developing countries, as well as a more detailed explanation of the panel threshold nonlinear model used in the study. Additionally, the abstract could benefit from more specific and concise language to improve clarity and understanding for readers.

3. Introduction looks adequate but following can be improved:

a. Providing a more detailed context for the global monetary crisis of 2007-2008 and its impact on the role of central banks in ensuring financial stability.

b. Citing more recent and relevant empirical studies on the relationship between CBI, MAPP, and financial stability, in addition to those mentioned in the introduction.

c. Providing a more specific and concise definition of the credit gap and its relevance to financial stability, to help readers better understand the study's main criteria for measuring stability.

d.Explaining the rationale for using a panel threshold nonlinear model and the dummy variable technique, as these are relatively technical and may require more explanation for readers who are not familiar with them.

3. Literature needs to be detailed. Very few studies.

4. No research questions or hypothesis ? provide them

5. Overall, the econometric methodology is appropriate for this research, but there may be some gaps in addressing potential heterogeneity across nations and providing a more detailed explanation of the non-standard poolability test. Why not fixed model ? Generalized Method of Moments (GMM)

6. The data collection and sources are adequate for this study. But provide more information on the selection of the 20 developing nations included in the analysis. Were they selected randomly or based on specific criteria? What was the rationale behind the selection?

7. Tests are adequate but no proper discussInstead of repeating the same information from earlier sections, the conclusion should provide a brief and straightforward overview of the study's key results and implications. The language could also be made more accessible to readers who may not have a background in econometrics. Additionally, the conclusion could suggest future research directions, potential limitations of the study, and areas where policymakers could focus their efforts based on the study's findings

8. Practical implications are missing.

7. PLOS authors have the option to publish the peer review history of their article (what does this mean?). If published, this will include your full peer review and any attached files.

Reviewer #1: No

Reviewer #2: No

While revising your submission, please upload your figure files to the Preflight Analysis and Conversion Engine (PACE) digital diagnostic tool, https://pacev2.apexcovantage.com/. PACE helps ensure that figures meet PLOS requirements. To use PACE, you must first register as a user. Registration is free. Then, login and navigate to the UPLOAD tab, where you will find detailed instructions on how to use the tool. If you encounter any issues or have any questions when using PACE, please email PLOS at figures@plos.org. Please note that Supporting Information files do not need this step.<quillbot-extension-portal></quillbot-extension-portal>

---

## [Author Response · Author response to Decision Letter 1]

27 Apr 2023

We have attached response to reviewers form

---

## [Decision Letter · Decision Letter 2]

2 May 2023

Evaluation of Central Bank Independence, Macroprudential Policy, and Credit Gap in Developing Countries

PONE-D-23-00739R2

Dear Dr. ANWAR,

We’re pleased to inform you that your manuscript has been judged scientifically suitable for publication and will be formally accepted for publication once it meets all outstanding technical requirements.

Kind regards,

Muhammad Kamran Khan, PhD Finance

Academic Editor

PLOS ONE

Additional Editor Comments (optional):

Reviewers' comments:

Reviewer's Responses to Questions

**Comments to the Author**

1. If the authors have adequately addressed your comments raised in a previous round of review and you feel that this manuscript is now acceptable for publication, you may indicate that here to bypass the “Comments to the Author” section, enter your conflict of interest statement in the “Confidential to Editor” section, and submit your "Accept" recommendation.

Reviewer #2: All comments have been addressed

2. Is the manuscript technically sound, and do the data support the conclusions?

Reviewer #2: Yes

3. Has the statistical analysis been performed appropriately and rigorously? 

Reviewer #2: Yes

4. Have the authors made all data underlying the findings in their manuscript fully available?

Reviewer #2: Yes

5. Is the manuscript presented in an intelligible fashion and written in standard English?

Reviewer #2: Yes

6. Review Comments to the Author

Reviewer #2: Thank you for resubmitting the paper. You have worked on addressing the previous comments. The paper is adequately supporting the information.

7. PLOS authors have the option to publish the peer review history of their article (what does this mean?). If published, this will include your full peer review and any attached files.

Reviewer #2: **Yes: **Dr. Muhammad Usman Tariq

<quillbot-extension-portal></quillbot-extension-portal>

---

## [Editor Report · Acceptance letter]

8 May 2023

PONE-D-23-00739R2 

Evaluation of Central Bank Independence, Macroprudential Policy, and Credit Gap in Developing Countries 

Dear Dr. ANWAR:

I'm pleased to inform you that your manuscript has been deemed suitable for publication in PLOS ONE. Congratulations! Your manuscript is now with our production department. 

Kind regards, 

on behalf of

Dr. Muhammad Kamran Khan 

Academic Editor

PLOS ONE